# UV/Vis-Based Persulphate Activation for *p*-Nitrophenol Degradation

**Valentin Dubois** [1,2]**, Carmen S. D. Rodrigues** [1,*]**, Ana S. P. Alves** [1] **and Luis M. Madeira** [1]

[1]    LEPABE—Laboratory for Process Engineering, Environment, Biotechnology and Energy, Faculty of Engineering, University of Porto, Rua Dr. Roberto Frias, 4200-465 Porto, Portugal; vdubois@etud.insa-toulouse.fr (V.D.); up201506349@fe.up.pt (A.S.P.A.); mmadeira@fe.up.pt (L.M.M.)

[2]    INSA—Institut National des Sciences Appliquées de Toulouse, 135, Avenue de Rangueil, 31400 Toulouse, France

[*]    Correspondence: csdr@fe.up.pt; Tel.: +351-22-4143654; Fax: +351-22-5081449

**Abstract:** In the present work, the degradation of *p*-nitrophenol (PNP) and its mineralization by a UV/Vis-based persulphate activation process was investigated. Firstly, a screening of processes as direct photolysis, persulphate alone and persulphate activated by radiation was performed. The incidence of radiation demonstrated to have an important role in the oxidant activation, allowing to achieve the highest PNP and total organic carbon (TOC) removals. The maximum PNP oxidation (100%) and mineralization (61.6%)—both after 2 h of reaction time—were reached when using T = 70 °C, $(S_2O_8^{2-})$ = 6.4 g/L and I = 500 W/m². The influence of radiation type (ultraviolet/visible, visible or simulated solar light) was also evaluated, being found that the source with the highest emission of ultraviolet radiation (UV/visible) allowed to achieve the best oxidation efficiency; however, solar radiation also reached very-good performance. According to quenching experiments, the sulphate radical is key in the activated persulphate oxidation process, but the hydroxyl radical also plays an important role.

**Keywords:** persulphate; activation by radiation; artificial light; simulated solar light; *p*-nitrophenol oxidation; mineralization



## 1. Introduction

*p*-Nitrophenol (PNP) has several industrial applications. Effectively, it is applied both as raw material and as a process intermediary in numerous types of industrial activities, such as the production of pharmaceutical and petrochemical products, explosives, pesticides, pigments, dyes, plasticisers, wood preservatives, and rubber chemicals, among others [1–3]. Due such widespread use, small concentrations of this chemical compound have been detected in both water resources and soils due to improper wastewater discharges resulting from production, distribution, and application processes [4–6]. This has alarmed the scientific community, as PNP is acutely toxic (the $LD_{50}$, in rats, is 250 mg/kg [7]), carcinogenic and recalcitrant, thus presenting harmful effects both on the environment and on human life [8–10]. As a consequence, PNP has been considered by the US Environmental Protection Agency (EPA) as a priority (micro)pollutant and its concentration is limited to less than 10 ng/L in natural waters [11–13]. Moreover, the maximum recommended concentration in drinking water is 60 μg/L [14].

Therefore, to minimize the adverse effects caused by this chemical, it is essential to treat wastewater before it is discharged into the water bodies. The conventional biological treatment is inefficient on its degradation due to the electro-withdrawing effect of the nitro group in microorganisms [1,2,15]. On the other hand, physical processes, like adsorption [16], only transfer the pollutant from one phase to another. Thus, it is crucial achieving an alternative solution that would enable to respond to this environmental challenge, ensuring the correct degradation of PNP.

Advanced oxidation processes (AOPs) are safe and effective techniques for removing resistant pollutants, maximising pollutants' degradation and/or mineralization, while producing nontoxic and environmentally friendly by-products like water and carbon dioxide [5]. Among AOPs, the persulphate-based oxidation stands out because the oxidant is more stable and cheaper than others (e.g., hydrogen peroxide [17]), can be applied in a wide range of pH [4,5], and generated sulphate radicals ($SO_4^{\bullet-}$), which have a redox potential ($E_0$ = 2.6 eV) similar to hydroxyl ($E_0$ = 2.8 eV), have a longer half-life time (3–4 $\times$ $10^{-5}$ s for $SO_4^{\bullet-}$ versus 2 $\times$ $10^{-8}$ s for $HO^{\bullet}$) [8,15,18–20].

The formation of the sulphate radical is achieved from persulphate activation that can be carried out by non-catalytic or catalytic methods, namely by heat, radiation or ultrasound—Equation (1), in the presence of transition metal ions ($Me^{n+}$)—Equation (2), and by pH change (imposing either alkaline or acidic conditions)—Equation (3) or Equations (4) and (5), respectively [1,6,21,22]. If heat, radiation, or ultrasound is used (Equation (1)), what happens is the cleavage of the oxygen-binding between the two sulphate groups [9,23,24]; these are the most efficient methods of activation, since one molecule of $S_2O_8^{2-}$ originates two of $SO_4^{\bullet-}$ [7,25]. If the presence of transition metal ions is used (Equation (2)), the formation of sulphate radicals occurs through the oxidation-reduction reaction of metal ions [18,26,27]; finally, if alkaline conditions are implemented (Equation (3)), radicals are formed due to the reaction that occurs between persulphate and $HO^-$ [3], while in acid medium the persulphate reacts with $H^+$ generating sulphate radicals (Equations (4) and (5)) [28]. These activation methods can be used individually or combined.

$$S_2O_8^{2-} \xrightarrow{\text{heat or radiation or ultrasound}} 2\,SO_4^{\bullet-} \tag{1}$$

$$S_2O_8^{2-} + Me^{n+} \rightarrow Me^{(n+1)} + SO_4^{\bullet-} + SO_4^{2-} \tag{2}$$

$$2S_2O_8^{2-} + 4HO^- \rightarrow 3SO_4^{2-} + SO_4^{\bullet-} + 2H_2O + O_2^{\bullet-} \tag{3}$$

$$S_2O_8^{2-} + H^+ \rightarrow HS_2O_8^- \tag{4}$$

$$HS_2O_8^- \rightarrow SO_4^{2-} + SO_4^{\bullet-} + H^+ \tag{5}$$

It should be noted that, under alkaline conditions, sulphate radicals can still form hydroxyl radicals (Equation (6)), both being responsible for the degradation of the organic pollutants [13,23,24,29]. Of course, depending on the pH, one species may be more dominant than another [21,25]. Typically, under acidic conditions, $SO_4^{\bullet-}$ is the predominant radical [5,19]; in alkaline conditions, is $HO^{\bullet}$ [3,6]. One feature of such AOP is that it allows the treatment of a wide variety of compounds, including toxic organic compounds, compounds mixtures or complex wastewater, because the generated sulphate and hydroxyl radicals are quite oxidizing and non-selective species.

$$SO_4^{\bullet-} + HO^- \rightarrow SO_4^{2-} + HO^{\bullet} \tag{6}$$

The activation of persulphate by radiation is promising when compared to the other methods [24,27] because it requires mild reaction conditions, has no metal ion leakage and has high efficiency in the degradation of refractory compounds [5,21,29]. Thus, it is beneficial to use radiation to activate the persulphate. The radiation-based persulphate activation process has already been studied to remove various pollutants or treat wastewater. Table 1 shows some studies reported in the literature, indicating the operating conditions used as well as the efficiency achieved.

**Table 1.** Studies focused on the radiation-activated persulphate process for pollutants removal or wastewater treatment.

| Pollutant/Wastewater | Operating Conditions | Removals | Reference |
|---|---|---|---|
| Methyl-tert-butyl ether (MTBE) | (MBTE) = 50 mg/L<br>$(S_2O_8^{2-})$ = 20 mg/L<br>$pH_{initial}$ = 7.0<br>UV-C radiation<br>Lamp power = 8 W<br>t = 200 min<br>I = 0.007 W/cm$^2$ | MTBE = 100%<br>TOC = 73.6% | [5] |
| 2,4-dinitroanisole (DNAN) | $(S_2O_8^{2-})$ = 10 mM<br>$pH_{initial}$ = 7.0<br>UV-C radiation<br>Lamp power = 15 W<br>Photon flux = $6.67 \times 10^{-8}$ E/(L.s)<br>t = 120 min<br>T = 20 °C | DNAN = 100% | [9] |
| Propyl paraben (PP) | $(S_2O_8^{2-})$ = 250 mg/L<br>[PP] = 200 mg/L<br>$pH_{initial}$ = 6.0<br>LED UV-A Lamp power = 10 W<br>Photon flux = $9.6 \times 10^{-7}$ E/(L.s)<br>T = 27 °C<br>t = 60 min | PP = 100% | [13] |
| Nanofiltration concentrated leachate | $(S_2O_8^{2-})$ = 18.0 g/L<br>$pH_{initial}$ = 9.0<br>UV Lamp power = 60 W<br>T = 80 °C<br>t = 8 h | COD = 65.4%<br>$NH_3$ = 51.4%<br>$UV_{254\ nm}$ = 98.1% | [19] |
| Bis(2-chloroethyl) ether (BCEE) | $(Na_2S_2O_8)$ = 15 mM<br>(BCEE) = 4 mg/L<br>$pH_{initial}$ = 2.0<br>Lamp power = 40 W<br>T = 25 °C<br>t = 60 min<br>UV-visible radiation | BCEE = 95%<br>TOC = 56% | [21] |
| Benzophenone-3 (BP-3) | BP-3:$S_2O_8^{2-}$ molar ratio = 1:500<br>$pH_{initial}$ = 7.0<br>UV radiation<br>Lamp power = 500 W<br>T = 40 °C<br>t = 650 min | BP-3 = 100% | [23] |
| Diatrizoate (DTZ) | (DTZ) = 30 mg/L<br>$(S_2O_8^{2-})$ = 24 mM<br>$pH_{initial}$ = 6.5<br>Simulated solar radiation with UV filter<br>Lamp power = 1.5 KW<br>T = 20 °C<br>t = 60 min | DTZ = ~83% | [25] |
| Phenacetin (PNT) | $(S_2O_8^{2-})$ = 0.3 mM<br>$pH_{initial}$ = 7.0<br>Simulated solar radiation<br>Lamp power = 500 W<br>Quantum yield = $1.14 \times 10^{-4}$ E/(m$^2$.s)<br>T = 27 °C<br>t = 10 min | PNT = 100% | [29] |

The application of persulphate-based oxidation to degrade PNP has been the subject of previous studies. Research works reported in the literature used several activation methods, such as heat, alkaline and/or dissolved metal [3,4,8,26], $CuFe_2O_4$ magnetic nano-particles [30], and different catalysts like Fe/Cu [2], silica gel/ Zn(0) [1], sulphur-doped ordered mesoporous carbon [31] or biochar [32] materials.

In this study, the degradation of PNP and its mineralization by UV/visible or visible radiation-based persulphate activation was performed. Firstly, the effect of persulphate concentration, radiation intensity and the radiation type was evaluated. Then, the predominance of the radical species (sulphate or hydroxyl) responsible for the oxidation process performance was assessed. Up to the authors' knowledge, this activation methodology was not addressed in the literature before for this pollutant degradation.

## 2. Results and Discussion

### 2.1. Comparison between Direct Photolysis, Persulphate and Persulphate Activated by Radiation

This study was started with the use of UV-visible radiation (TQ150 lamp) and it was firstly aimed to compare the performance of direct photolysis (only radiation, no persulphate), persulphate per se (no radiation) and persulphate activated with the artificial radiation. The efficiency of all processes were assessed in terms of PNP and TOC removals, being the results obtained shown in Figure 1. PNP absorbs in the 200–500 nm UV/visible range (absorption spectrum not shown), which practically covers all emission wavelengths of the TQ150 lamp (200–600 nm). However, direct photolysis proved to be inefficient in reducing the pollutant content and mineralising the organic matter, with overall removals below 8% after 2 h of irradiation, which allows concluding that PNP is not very sensitive to UV/visible radiation. This low efficiency of the process is corroborated by the fact that there is only a slight decrease of the medium pH (Figure 1d), as a consequence of carboxylic acids formation. The persulphate alone leads to an increase in PNP (38.5%) and TOC (16.9%) removals (Figure 1, being the data performances indicated reached after 2 h), as well as to a more pronounced decrease of pH (Figure 1d). In this case, the pH decrease is associated to the formation of $H^+$ during the persulphate process, which was corroborated by the fact this parameter also reduced in the blank run (without PNP), but also to the generation of intermediate compounds like carboxyl acids. This improved efficiency of the process is due to the formation of sulphate radical ($SO_4^{\bullet-}$), responsible for the PNP and intermediate compounds oxidation, by activation of persulphate in the presence of $H^+$ (Equations (4) and (5)); this is corroborated by the observed decrease in persulphate concentration along reaction time (see Figure 1c).

In turn, the activation by the acid pH and radiation together improved the efficiency of the process, reaching, after 2 h, the highest overall PNP (70.2%) and TOC (41.7%) removals and the greatest reduction of persulphate concentration and pH. This is due to the fact that more sulphate radicals are available for the reaction with both PNP and intermediate compounds formed as a consequence of the additional oxidant activation via UV/visible radiation (Equation (1)).

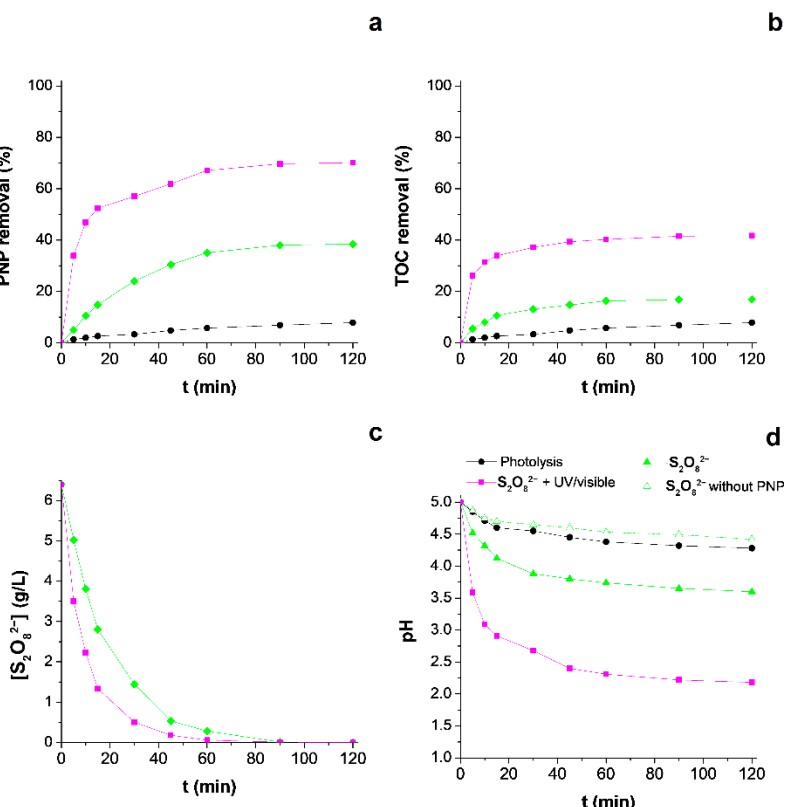

**Figure 1.** Evolution of *p*-nitrophenol (PNP) (**a**) and total organic carbon (TOC) (**b**) removals, concentration of persulphate remaining in solution (**c**) and pH (**d**) during photolysis, persulphate alone and persulphate activated with UV/visible light ([PNP]$_\text{o when used}$ = 500 mg/L, pH = 5.0, T = 30 °C, I$_\text{when used}$ = 500 W/m$^2$ and [S$_2$O$_8$$^{2-}$]$_\text{o,when used}$ = 6.4 g/L).

The results obtained are in accordance with those reported by He et al. [19], who observed the same tendency when treating a concentrated leachate. On the other hand, Ioannidi et al. [13] observed that the degradation of propylparaben by direct photolysis is insignificant (only 1% was reached after 60 min of the oxidative process) and an improvement in the pollutant removal was achieved when the persulphate was activated by UV-A radiation. Gao et al. [33] report lower oxidation of sulfamethazine (15.1–22.0% within 45 min), as well as apparent kinetic constant ($3.5 \times 10^{-3}$–$5.6 \times 10^{-3}$ min$^{-1}$) for persulphate per se and photolysis. However, the combination of persulphate with UV radiation allowed to increase the process efficiency, leading to an increase in the removal to 96.5% and the apparent kinetic constant raised more than an order of magnitude, to $7.5 \times 10^{-2}$ min$^{-1}$. In a study that evaluated the degradation of methyl paraben, the use of persulphate alone has shown low efficiency in pollutant degradation (10% of removal, after 90 min, and a degradation rate constant of 0.001 min$^{-1}$) [34]; an improvement of the contaminant oxidation was observed when applying UV-C radiation (34% and rate constant of 0.005 min$^{-1}$), this increase being due to the fact that the mercury lamp emits radiation in wavelength (256 nm) close to the maximum absorption band of methyl paraben; even more, the maximum oxidation (98.9%) and degradation rate constant (0.0396 min$^{-1}$) were achieved when using the persulphate and UV-C radiation simultaneously. Zhou et al. [25] also found that diatrizoate does not degrade by the incidence of radiation or presence of persulphate. However, its removal is achieved when the oxidant is combined with radiation, being achieved a reduction of the pollutant concentration of 42% after 60 min of reaction. Finally, Tan el al. [29] found low removals of phenacetin at pH 5.5, 7.0 and 8.5 (13, 26.3 and 25.8%, respectively, after a reaction time of 30 min) in the presence of solar radiation, and negligible degradation of the pollutant by persulphate alone, however the

combination of persulphate with radiation leads to a complete oxidation of phenacetin for the three pH levels studied.

## 2.2. Effect of the Temperature

In order to evaluate the effect of the reaction temperature on the process efficiency, four runs were carried out where this parameter was changed between 30 to 90 °C. As shown in Figure 2 and Table 2, the PNP removal and organics mineralization, as well as the reduction of pH, increased with the temperature up to 70 °C. This is due to the fact that the reaction kinetic constants for both the formation of sulphate radicals (e.g., through thermal activation of persulphate, Equation (1)) and for organics degradation increase with temperature according to Arrhenius' law. However, a slight decrease in the efficiency of the process was observed for the maximum temperature tested, which is associated with the reduction of the amount of sulphate radicals available to degrade the PNP and the intermediates formed at 90 °C once, in a short reaction period, the number of radicals generated is so high that the undesired parallel reaction between them (Equation (7)) is accelerated [35,36]—the so-called scavenging effect. Thus, the best temperature selected for subsequent tests was 70 °C.

$$SO_4^{\bullet -} + SO_4^{\bullet -} \rightarrow S_2O_8^{2-} \tag{7}$$

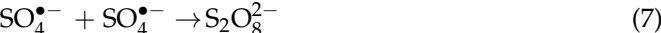

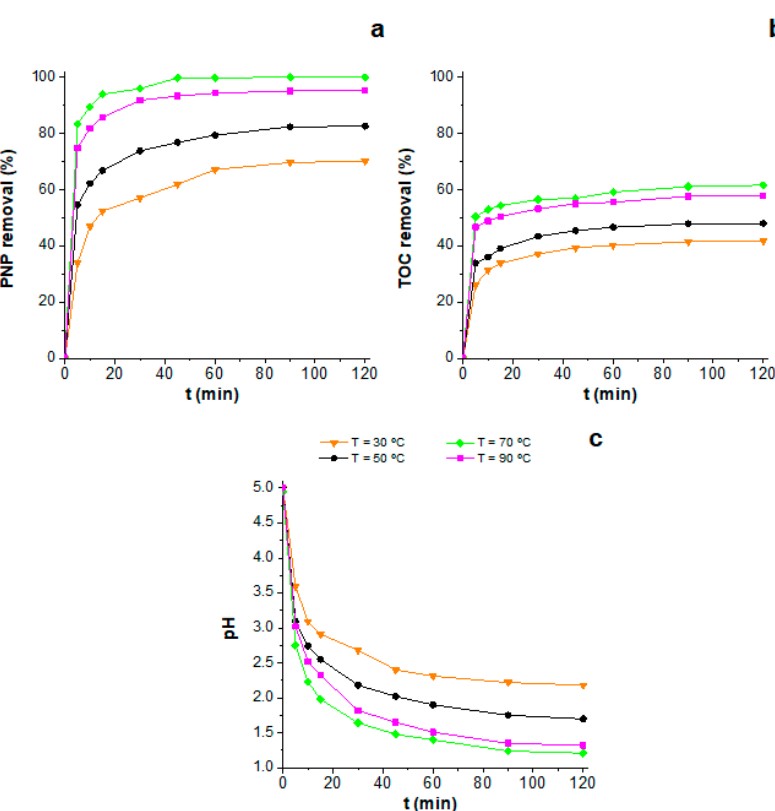

**Figure 2.** Influence of temperature on PNP (**a**) and TOC (**b**) removals and pH evolution (**c**) during the persulphate activated by radiation process ([PNP]$_o$ = 500 mg/L, pH = 5.0, [S$_2$O$_8^{2-}$]$_o$ = 6.4 g/L and I = 500 W/m$^2$).

**Table 2.** Conditions employed in the activated persulphate process and PNP and TOC removals after 2 h. In bold are highlighted the optimum conditions in each set of the parametric study.

| Parameter Assessed | T (°C) | $[S_2O_8^{2-}]$ (g/L) | I (W/m$^2$) | Radiation Type | Removal (%) | |
|---|---|---|---|---|---|---|
| | | | | | PNP | TOC |
| Temperature | 30 | 6.4 | 500 | UV/visible | 70.2 | 41.7 |
| | 50 | | | | 82.6 | 48.0 |
| | **70** | | | | **100.0** | **61.6** |
| | 90 | | | | 95.4 | 57.7 |
| Persulphate concentration | 70 | 0 | 500 | UV/visible | 7.8 | 7.2 |
| | | 0.8 | | | 41.3 | 32.1 |
| | | 1.6 | | | 56.6 | 46.0 |
| | | 3.2 | | | 73.5 | 53.7 |
| | | **6.4** | | | **100.0** | **61.6** |
| | | 8.0 | | | 89.2 | 57.3 |
| Irradiation intensity | 70 | 6.4 | 0 | UV/visible | 79.8 | 45.5 |
| | | | 100 | | 82.1 | 48.0 |
| | | | 150 | | 86.9 | 50.8 |
| | | | 250 | | 93.4 | 53.2 |
| | | | 449 | | 94.1 | 57.1 |
| | | | **500** | | **100.0** | **61.6** |
| Radiation type | 70 | 6.4 | 449 | Without Radiation | 79.8 | 45.5 |
| | | | | Visible | 82.2 | 48.0 |
| | | | | Simulated Solar | 87.6 | 50.1 |
| | | | | **UV/Visible** | **94.1** | **57.1** |

The positive influence of the temperature on the efficiency of persulphate activated by radiation was also reported in the literature. Pan et al. [23] observed an improvement of abatement of benzophenone-3 from 11% to 100%, after 3 h of reaction, when increasing the temperature from 25 to 40 °C; inherently, the pseudo-first-order rate constant also increased. The authors associate the excellent performance of the process at 40 °C with the high solubility of the pollutant to be removed and the formation of more active sulphate and hydroxyl radicals at this temperature [23]. He et al. [19], when treating a nanofiltration concentrated leachate by the UV/persulphate process, reported an increase of aromatic compounds and ammonia nitrogen removals when temperature increased from 50 to 80 °C, but no further improvement was observed at 90 °C. On the other hand, COD removal increased with temperature in the whole range tested (50–90 °C).

### 2.3. Influence of Persulphate Concentration

The dose of oxidant is an important parameter to be taken into account in the optimization of any AOP since it has a high contribution in the cost of the treatment process. For this purpose, five experiments were carried out, being the persulphate concentration varied in the range from 0.8 g/L to 8.0 g/L (corresponding to 0.1 and 1.0 times the stoichiometric amount of persulphate required for total elimination of the chemical oxygen demand (COD)—12 $g_{persulphate}/g_{COD}$ [37]).

Figure 3 shows the evolution of PNP and TOC removals (as well as the pH profile) during the reaction time for the different levels of oxidant employed; for reference, the direct photolysis experiment was also included (blank run, wherein $[S_2O_8^{2-}]$ = 0 g/L). The abatement of the pollutant and the organics mineralization increased with the oxidant dose up to 6.4 g/L, reaching, after 2 h of reaction, removals of 100% and 61.6%, respectively (see Table 2). Again, and as shown in previous figures, very similar patterns are observed in PNP and TOC removals and the pH decay with the parameter under study. In this case, pH has reached the greatest decay (from 5.0 to 1.2) for the persulphate concentration of 6.4 g/L, which indicates a greater formation of carboxylic acids for this dose of oxidant.

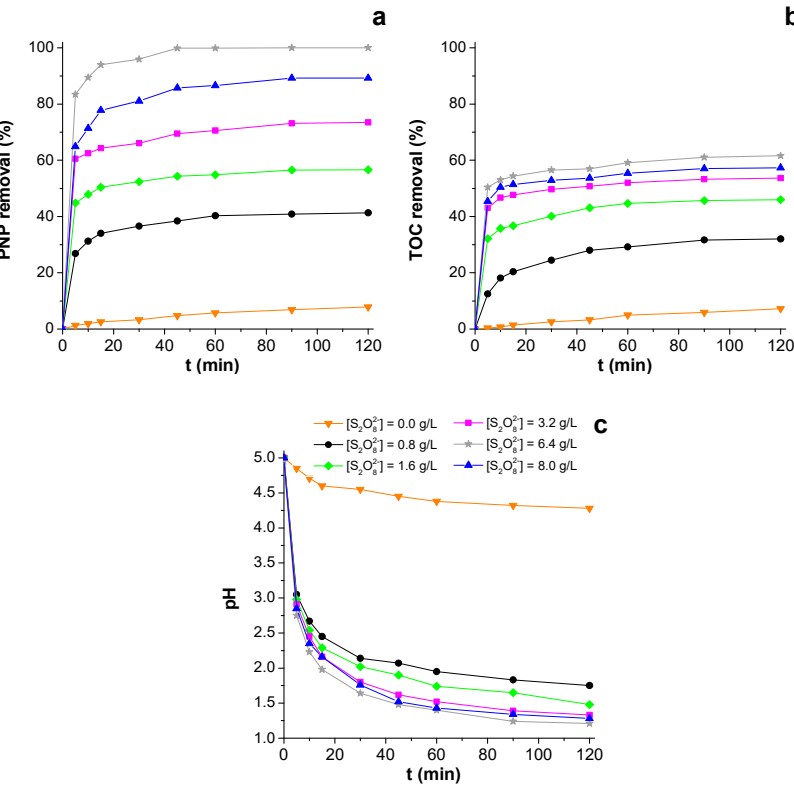

**Figure 3.** Effect of persulphate concentration on PNP (**a**) and TOC (**b**) removals and pH evolution (**c**) during the oxidation process ([PNP]$_o$ = 500 mg/L, pH = 5.0, T = 70 °C and I = 500 W/m$^2$).

The results found are in agreement with those reported in the literature. For instance, for the treatment of a concentrated leachate, He et al. [19] reached the maximum removal of COD, ammonia nitrogen and aromatic compounds with a concentration of persulphate of 18 g/L, but the process efficiency decays for higher doses of the oxidant. In another study where it was evaluated the degradation of tetramethylammonium hydroxide, the rate constant increased with the persulphate concentration from 10 to 50 mM, but the authors observed a slightly negative effect for higher oxidant doses [38].

The appearance of an optimal concentration of persulphate can be explained by the scavenging of sulphate radicals in the presence of an excess of oxidant (Equation (8)); although the $S_2O_8^{\bullet-}$ radical is formed, it has a lower oxidation potential compared to sulphate. Moreover, when increasing the concentration of the persulphate radical present in solution (as a consequence of the increased oxidant dose), a recombination of radicals occurs according to Equation (7) [35]. Therefore, the amount of these radicals available for the degradation of PNP and/or intermediate compounds decreases.

$$S_2O_8^{2-} + SO_4^{\bullet-} \rightarrow SO_4^{2-} + S_2O_8^{\bullet-} \tag{8}$$

In contrast, Ioannidi et al. [13] observed an increase in the propyl paraben degradation (from 27.6 to 100%, after 30 min), as well as in the apparent kinetic constant (from ~0.01 to ~0.17 min$^{-1}$), when the concentration of persulphate increased from 100 to 500 mg/L. Still, in the removal of propyl paraben, Zhou et al. [25] reported the same tendency, i.e., the pollutant oxidation performance and apparent kinetic constant increase with persulphate concentration in the range 6 to 24 mM. The authors state that the improvement in the process' performance is associated with the fact that no radical scavenging reaction (Equation (8)) occurs in the persulphate concentration range tested. The same tendency and justification was reported in the studies developed by Sakulthaew et al. [39] and Dhaka et al. [34] when degrading 17β-estradiol and methyl paraben, respectively, by UV/persulphate.

### 2.4. Effect of the Irradiation Intensity

The radiation intensity has an important role in the efficiency of the photo-assisted oxidation process. To evaluate this parameter's effect, experiments were carried out where the intensity varied up to $500 \, W/m^2$. Apart from the control run, without radiation, the two lowest intensities (100 and $150 \, W/m^2$) correspond to the minimum and the maximum solar radiation incidence in the north of Portugal [40]. However, higher intensities, including the maximum emitted by the TQ150 lamp ($500 \, W/m^2$), were tested in order to simulate the results to other regions/countries where the incidence of solar radiation is larger.

The PNP removal, mineralization degree and the decay in pH are more pronounced with the light intensity increase (see Figure 4 and Table 2), as expected. This increase in the oxidative process efficiency is related to the improvement of the sulphate radicals' formation, according to Equation (1). These results are in agreement with other studies reported in literature, namely those which applied persulphate activated by ultraviolet light in the degradation of 17β-estradiol [39] and tetramethylammonium hydroxide [38]. The authors observed an increased in the rate constants when improving the lamp power from 4 to 8 W and 8 to 15 W, respectively.

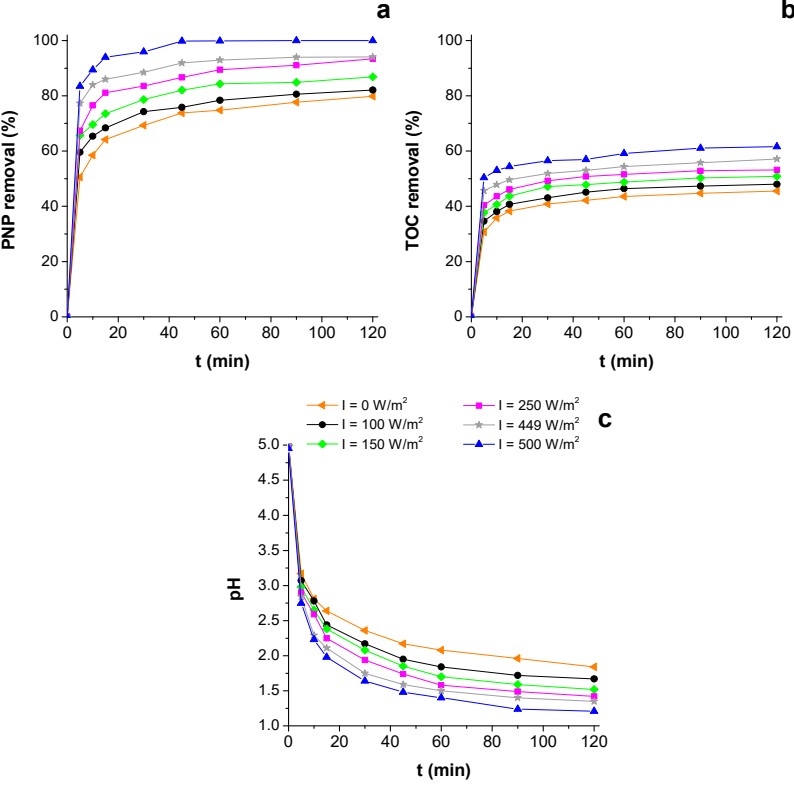

**Figure 4.** PNP (**a**) and TOC (**b**) removals and pH evolution (**c**) for different radiation intensities during the persulphate oxidation process ($[PNP]_o = 500 \, mg/L$, pH = 5.0, T = 70 °C and $[S_2O_8^{2-}]_o, = 6.4 \, g/L$).

### 2.5. Effect of Irradiation Type

Experiments were also carried out with UV/visible, visible only and simulated solar radiation to determine the effects related to the type of radiation on PNP abatement and organics mineralization. For a proper comparison of these radiation types, the runs were performed under the same conditions of irradiance; so, it was necessary to reduce the intensity of radiation in the UV/visible lamp to $449 \, W/m^2$ by recirculating a Solophenyl BLE 155% dye solution on the water jacket quartz tube, as detailed in the Materials and Methods section.

Figure 5 shows that visible radiation is not very effective on persulphate activation, whereby the removals reached (82.2 for PNP and 48.0% for TOC after 2 h—Table 2) are very

close to the performances reached without radiation. However, when applying simulated solar radiation the process efficiency is much better; improvements are even higher when UV/visible radiation is used because the TQ150 lamp emits more UV radiation than the Xenon lamp used to simulate solar radiation. Therefore, the results show the importance of using UV radiation. Nevertheless, the possibility of using solar radiation is shown and shall be considered, once the energy cost is null. The results obtained are in line with was reported in other studies that evaluated the influence of UV, visible, and/or solar radiation in the degradation of other pollutants, namely propyl paraben [13] and benzophenone-3 [23].

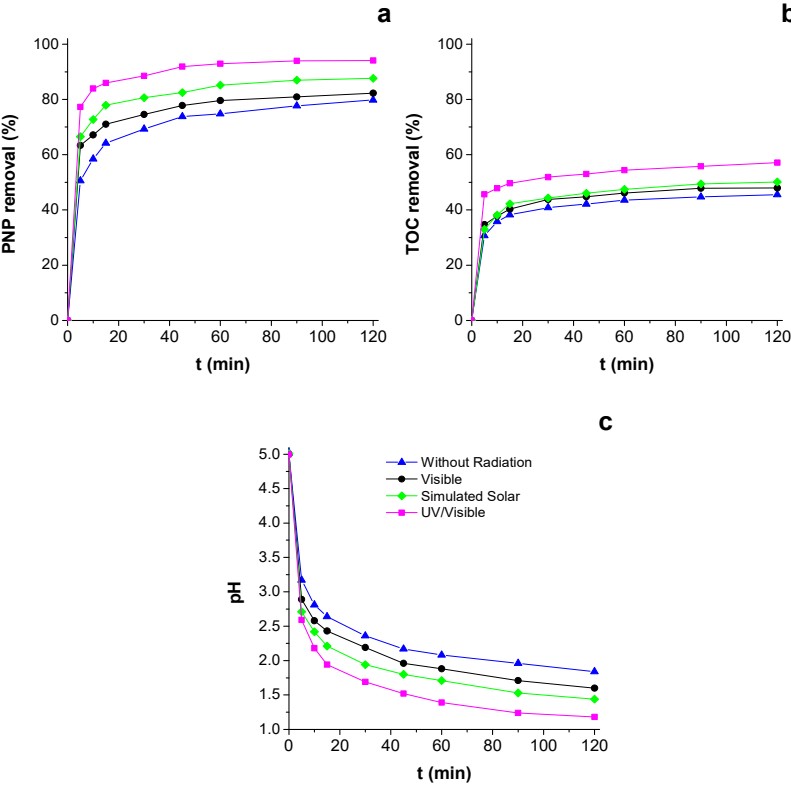

**Figure 5.** PNP (**a**) and TOC (**b**) removals and pH evolution (**c**) during the persulphate oxidation process when using different types of light ([PNP]$_o$ = 500 mg/L, pH = 5.0, T = 70 °C, I = 449 W/m$^2$ and [S$_2$O$_8$$^{2-}$]$_o$ = 6.4 g/L).

*2.6. Radicals Role*

Using the best conditions reported so far, two new experiments were carried out in the presence of ethanol (EtOH) or *tert*-butyl alcohol (TBA) with 2.4 M concentration of the scavenger (corresponding to a persulphate:scavenger molar ratio of 1.7, as proposed by other authors [41]) to assess the presence of sulphate and hydroxyl radicals and their contribution to the oxidation performances. The results obtained are compared in Figure 6 with those obtained without the addition of scavengers. The two scavengers' selection took into account the fact that the first (EtOH) reacts with both radicals with a similar rate of reaction, while the second (TBA) reacts at a higher rate with the hydroxyl as compared to the sulphate radical [25,42].

Figure 6 shows the effect of using these quenchers on PNP abatement, TOC removal and reduction of solution pH during the persulphate activated by UV/visible radiation process. It was found that both scavengers lead to a reduction in the oxidative process efficiency, as well as in decay of pH. However, this effect is more pronounced in the presence of EtOH than TBA. Thus, it is concluded that the sulphate radical dominated the oxidative process, but a contribution of the hydroxyl radical in the degradation of the contaminant and organics mineralization also occurs.

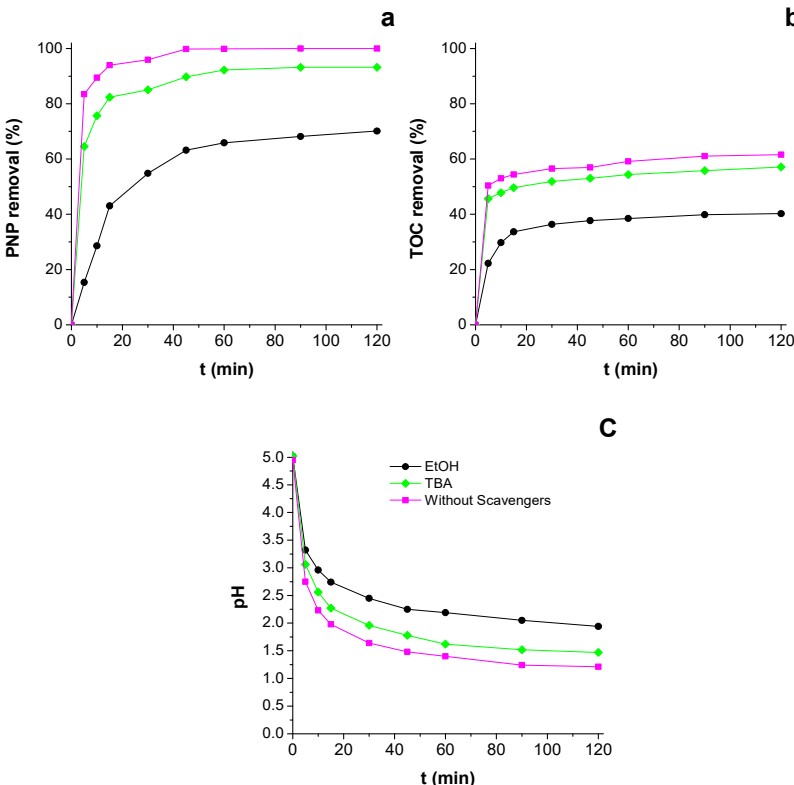

**Figure 6.** PNP (**a**) and TOC (**b**) removals and pH evolution (**c**) during the persulphate oxidation process activated by radiation in presence or not of scavengers ([PNP]$_o$ = 500 mg/L, pH = 5.0, T = 70 °C, I = 500 W/m$^2$, [S$_2$O$_8^{2-}$]$_o$ = 6.4 g/L and (persulphate:scavenger) $_{o,when\ used}$ = 1.7 mol/mol).

Dhaka et al. [34] also verified, under acid and neutral pH conditions, a higher reduction of methyl paraben removal in presence of EtOH (decay from 100% to 54.3–52.7%) than with TBA (from 100% to 80.2–77.1%). Still, Zhou et al. [21] observed, at pH 3 and 7.0, a much greater decrease of reduction of bis(2-cloroethyl) ether, after 60 min of reaction, in presence of EtOH (decrease from 90.7 to 53.6% and from ~73 to 46.4%, respectively) than with TBA (decrease from 90.7 to 88.7% at pH = 3.0 and, from ~73 to ~62% at pH = 7.0); however, in alkaline conditions (pH = 10) the decay of the pollutant removal also occurs in the presence of both radical quenchers but was higher in presence of TBA than EtOH (from ~83 to ~46 and ~66%, respectively). The authors concluded that in acid and neutral conditions the sulphate radical dominated, but in alkaline medium is the hydroxyl radical that prevails.

## 3. Materials and Methods

### 3.1. Chemicals

In this study PNP (C$_6$H$_5$NO$_4$, which has a molecular weight of 155.11 g/mol) was used as a model compound, being purchased from Acros Organics with a purity of 99%. The oxidant (potassium persulphate) was obtained from Alfa Aesar with a purity of 99%. The scavengers ethanol—EtOH—(≥96%) and *tert*-butyl alcohol—TBA—(99.5%) were from VWR Chemicals and Sigma-Aldrich, respectively.

### 3.2. Experimental Procedure

The degradation of PNP by persulphate activated by artificial light was carried out in a glass batch reactor with 500 mL-capacity. The photoreactor is equipped with a UV/visible Heraeus TQ 150 lamp (with a potency of 150 W, which corresponds to an intensity of 500 W/m$^2$, emitting radiation at wavelengths between 200 and 600 nm) that was axially located inside a submerged quartz tube. This tube has a jacket for water recirculation

connected to a thermostat (Hubber, polystat cc1) to keep the temperature inside the reactor constant ($\pm 1.0\,°C$).

The reactor was filled with 250 mL of a PNP solution (without adjusting the pH, i.e., using the natural pH of the solution—5.0) with a concentration of 500 mg/L (this concentration was typically found in real wastewaters [2,43]). After the solution has reached the desired temperature, the lamp was switched on and the irradiation emission allowed to stabilize (only 1.5 min are required—data not shown). Then, the desired amount of solid persulphate was added; this moment corresponded to the initial instant of the reaction ($t = 0$ min). The pH was fixed in the above value, which was the best condition achieved in a previous study wherein the degradation of the same pollutant by persulphate activated with alkaline conditions, temperature and iron was evaluated [3].

Throughout the runs, the solution inside the photoreactor was stirred at 200 rpm. For this purpose, a magnetic bar and a magnetic stirrer (VWR, model VS-C7) were used. During the reaction, samples were taken at specific reaction times (5, 10, 15, 30, 45, 60 and 120 min) to measure the concentration of PNP, total organic carbon (TOC) and pH.

The experiments wherein the radiation intensity's effect was evaluated were performed as described previously, i.e., by simple recirculation in the jacket quartz tube of a Solophenyl Green BLE 155% dye solution with a known concentration; more details are described elsewhere [44,45].

The run in which only visible radiation was employed was carried out in the same way as described above, but the quartz tube was substituted by a tube made of Duran glass, which filters the UV radiation.

The run with simulated solar radiation was performed in a Solarbox 3000e from N-Wissen GmbH equipped with a xenon lamp of 1500 W that emits radiation at wavelengths from 290 to 800 nm. The installation consists of two reactors with 250 mL of capacity. One reactor is placed outside the Solarbox being equipped with a jacket for water recirculation from a thermostatic bath (Hubber, polystat cc1) for temperature control; the other, without a jacket, was located inside the Suntest. Each reactor was fed with 125 mL of PNP solution (to keep the total of 250 mL as before) that was continuously recirculated, at a flow rate of 0.2 L/min, between them, using two peristaltic pumps (Miniplus3 from Gilson). After reaching the desired temperature, the lamp was switched on and the potassium persulphate added to the reactor located outside the box. During the reaction the contents of these reactors were stirred using a magnetic bar and stir plate (VWR, model VS-C7).

### 3.3. Analytical Methods

The concentration of PNP was determined by high performance liquid chromatography (HPLC) in a Hitachi Elite LaChrom apparatus. The chromatograph is equipped with a UV detector and uses a Purospher STAR RP-18 column (5 μm, 250 × 4.0 mm) that was heated up to 35 °C. An isocratic method was used with a flow rate of 0.75 mL/min of eluent composed by 90% water and 10% methanol. The automatic injection volume was 20 μL, and the detection was carried out by UV absorbance at a wavelength of 285 nm.

Total Organic Carbon (TOC), which is the difference between Total Carbon (TC) and Inorganic Carbon (IC), was measured on a Shimadzu TOC-L apparatus equipped with a Shimadzu SHL Autosampler. This parameter was measured according to the method 5310D [46]. For this a catalytic oxidation was conducted at 720 °C, followed by quantification of the $CO_2$ formed in an infra-red detector.

The concentration of persulphate remaining in solution was determined by measuring the absorbance at 352 nm that results from the reaction of persulphate with iodide in the presence of sodium bicarbonate [47]. For this purpose, a Thermo Electron Corporation Heilos $\gamma$ spectrophotometer was used.

All determinations were performed in duplicate, and the coefficient of variations were less than 2% for all parameters measured.

## 4. Conclusions

In this work it was evaluated the degradation of a priority pollutant (PNP) and its mineralization by persulphate activated by radiation. Firstly, the screening of the processes (photolysis, persulphate alone and combination of persulphate with UV/visible radiation) was assessed, allowing to conclude that the combination of radiation and oxidant permitted to achieve the maximum efficiency of the oxidative process. The direct photolysis contribution is low, <10% after 2 h, but the persulphate activation induced by the natural pH of the medium has a considerable contribution in the overall process efficiency, being more relevant, particularly in terms of mineralization, the radiation-activated part. The effect of the temperature, persulphate concentration, radiation intensity and the radiation type was also assessed; it was found that complete degradation of PNP and considerable TOC removal (61.6%) were reached in 2 h when applying the best operating conditions (T = 70 °C, $[S_2O_8^{2-}]$ = 6.4 g/L and I = 500 W/m$^2$ of UV/visible radiation). Nevertheless, the process is clearly feasible with solar radiation, opening the door for the industrial-scale implementation of the technology. According to the quenching experiments, both sulphate and hydroxyl radicals are responsible for the oxidation of PNP and intermediate compounds, but the sulphate radical dominates in the oxidative process.

Taking into account the outstanding performance achieved with persulphate activated by radiation, it is possible to conclude that this advanced oxidation process is a promising technology to be adopted in the treatment of effluents containing PNP, without the need of resorting to metal species as sulphate activators.

**Author Contributions:** V.D., validation and investigation; C.S.D.R., conceptualization, supervision, validation, investigation and writing—original draft; A.S.P.A., investigation and writing original draft; L.M.M., conceptualization, supervision, funding acquisition, resources and writing review and editing. All authors have read and agreed to the published version of the manuscript.

**Funding:** This work was financially supported by: Base Funding—UIDB/00511/2020 of the Laboratory for Process Engineering, Environment, Biotechnology and Energy—LEPABE—funded by national funds through FCT/MCTES (PIDDAC).

**Data Availability Statement:** Data sharing is not applicable to this article.

**Acknowledgments:** Valentin Dubois would like to gratefully acknowledge the Programme Erasmus+ Placements for funding his internship while Carmen Rodrigues thanks the Portuguese Foundation for Science and Technology (FCT) for the financial support of her work contract through the Scientific Employment Support Program (Norma Transitória DL 57/2017). The authors are grateful to Professor Joaquim Faria and Doctor Maria José Sampaio of Associate Laboratory LSRE-LCM for supplying the Solarbox.

**Conflicts of Interest:** The authors declare no conflict of interest.

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
