# Peer review of "UV/Vis-Based Persulphate Activation for p-Nitrophenol Degradation"

_catalysts, doi:10.3390/catal11040480_

Round 1

Reviewer 1 Report

In the manuscript „UV/VI-based Persulfate Activation for p-Nitrophenol Degradation“ the authors report a method for the removal of p-nitrophenol from waste water. By irradiation of UV light, persulfate is homolytically cleaved leading to oxidative degradation of the phenol. Reaction parameter were optimized to achieve full conversion.

The degradation of p-nitrophenol in waste water is an industrially important task. The procedure is operationally simple. Also sun light irradiation is sufficient. The paper is well written and the experimental results and details fully and reproducibly described. However, publication in “catalysts” seems not suited since the focus of the journal is on catalysts and catalytic transformations. This is not the case here, so transferal to a more appropriate journal is recommended.

In order to improve the quality of the paper further, the authors should address the following issues:

- The abstract is too long and should be shortened.

- In section 2.4. the irradiation intensity should be discussed with respect to the experimental setup. p-Nitrophenol shows strong absorption. How deep is the UV-light penetration into the solution?

- In section 3.2. (line 308) it should be clarified whether the persulfate is added as solid or solution.

Reviewer 2 Report

In this work  the degradation of p-nitrophenol and its mineralisation by a  
UV/Vis-activated persulfate was investigated.
The screening of the processes allowed to assess the combination of radiation and oxidant that achieve the maximum efficiency of the oxidative process.
An importal finding was that the process is clearly feasible with solar radiation, which can make possible a future industrial implementation.

The introduction provide a good view of the background, the references are appropriate in number and relevance, the reserch design is appropriate, methods and results are clearly described and appropriate, the conclusions are consistent.

The scope of this study is clear, even is somehow limited. The prominence of this research and therefore the interest for the average reader of the Catalysts may be not so high, but I think that this is anyway a valuable contribute to the knowledge in the field, and the solar radiation feasibility is interesting and promising. So, all considered, I recomend its publication in your Journal.

Suggested small improvements:
- radicals in (1)-(6) and (8) should be more visible, as in (7)
- alignment of (4)-(5) is confusing
- Figure 1c: differentiate colors for the two green lines
- the symbol for Celsius degrees should not have the underlining under the ° it's not the ordinal indicator
- you generally use the capital L for liters, but there are a couple of instances of "ml" instead.

Reviewer 3 Report

The ms of Rodrigues and co-workers is on the UV/Vis-based Persulfate Activation for p-Nitrophenol Degradation.

The research can be useful for the efficient elimination of p-nitrophenol from wastewater. However, a few modifications should be made in the ms:

- An English language check is required throughout the ms. It contains a few grammatical errors and bad wording.

- The au-s systematically mix the Introduction part with the Results and discussion. A new section should be made in the Introduction part discussing the main results of previous papers. The Results and discussion part should be dedicated to the new results only.

- The au-s use only diagrams in the ms. The most important results of the optimization should be listed in tables for better understanding. It would be much more useful for the readers.

The au-s should answer the following questions:

- Can this method be used for the degradation of other toxic compounds or compound mixtures?

- Is there any remaining persulfate after the degradation, if yes, what happens with it?

In conclusion, acceptance of this ms in Catalysts is suggested after the above revision.

Round 2

Reviewer 1 Report

In the revised version of the manuscript the authors have addressed all issues raised in the first review and made the corresponding corrections. The paper is therefore recommended for publication.